

# Characterisation and improvement of $j(\mathrm{O}^1\mathrm{D})$ filter radiometers

Birger Bohn[1], Dwayne E. Heard[2,3], Nikolaos Mihalopoulos[4,5], Christian Plass-Dülmer[6], Rainer Schmitt[7], and Lisa K. Whalley[2,3]

[1]Institut für Energie und Klimaforschung IEK-8, Forschungszentrum Jülich, 52428 Jülich, Germany
[2]School of Chemistry, University of Leeds, Leeds, LS2 9JT, UK
[3]National Centre for Atmospheric Science, University of Leeds, Leeds, LS2 9JT, UK
[4]Department of Chemistry, University of Crete, Heraklion 71003, Greece
[5]Institute for Environmental Research and Sustainable Development, National Observatory of Athens, Athens 11810, Greece
[6]Deutscher Wetterdienst, Meteorologisches Observatorium Hohenpeissenberg, 82383 Hohenpeissenberg, Germany
[7]Meteorologie Consult GmbH, Frankfurter Str. 28, 61462 Königstein, Germany

*Correspondence to:* B. Bohn
(b.bohn@fz-juelich.de)

**Abstract.** Atmospheric $\mathrm{O}_3 \rightarrow \mathrm{O}(^1\mathrm{D})$ photolysis frequencies $j(\mathrm{O}^1\mathrm{D})$ are crucial parameters for atmospheric photochemistry because of their importance for primary OH formation. Filter radiometers have been used for many years for in-situ field measurements of $j(\mathrm{O}^1\mathrm{D})$. Typically the relationship between the output of the instruments and $j(\mathrm{O}^1\mathrm{D})$ is non-linear because of changes in the shape of the solar spectrum dependent on solar zenith angles and total ozone columns. These non-linearities can be compensated by a correction method based on laboratory measurements of the spectral sensitivity of the filter radiometer and simulated solar actinic flux density spectra. Although this correction is routinely applied, the results of a previous field comparison study of several filter radiometers revealed that some corrections were inadequate. In this work the spectral characterisations of seven instruments were revised and the correction procedures were updated and harmonized considering recent recommendations of absorption cross sections and quantum yields of the photolysis process $\mathrm{O}_3 \rightarrow \mathrm{O}(^1\mathrm{D})$. Previous inconsistencies were largely removed using these procedures. In addition, optical interference filters were replaced to improve the spectral properties of the instruments. Successive determinations of spectral sensitivities and field comparisons of the modified instruments with a spectroradiometer reference confirmed the improved performance. Overall, filter radiometers remain a low-maintenance alternative of spectroradiometers for accurate measurements of $j(\mathrm{O}^1\mathrm{D})$ provided their spectral properties are known and potential drifts in sensitivities are monitored by regular calibrations with standard lamps or reference instruments.

## 1 Introduction

Atmospheric photolysis frequencies $j(\mathrm{O}^1\mathrm{D})$ are first-order rate constants that determine the formation rate of electronically excited $\mathrm{O}(^1\mathrm{D})$ atoms in the photolysis of ozone:

$$\mathrm{O}_3 + h\nu(\lambda \leq 340\,\mathrm{nm}) \longrightarrow \mathrm{O}(^1\mathrm{D}) + \mathrm{O}_2 \tag{R1}$$

$$\mathrm{d}[\mathrm{O}^1\mathrm{D}]/\mathrm{d}t = [\mathrm{O}_3] \times j(\mathrm{O}^1\mathrm{D}) - L[\mathrm{O}^1\mathrm{D}] \tag{1}$$



[O$^1$D] and [O$_3$] denote gas-phase concentrations of O($^1$D) and O$_3$, respectively. $L$ is the first-order total loss rate constant of O($^1$D) by quenching and chemical reactions, mainly by N$_2$, O$_2$ and H$_2$O. Because these loss processes are fast, very small steady-state concentrations $\ll 1\,\mathrm{cm}^{-3}$ are resulting for O($^1$D) even around noontime:

$$[\mathrm{O^1D}] \approx [\mathrm{O_3}] \times j(\mathrm{O^1D})/L \tag{2}$$

Nevertheless, O($^1$D) is extremely important for atmospheric chemistry because it can form OH radicals in a reaction with water vapour:

$$\mathrm{O(^1D) + H_2O \longrightarrow 2\,OH} \tag{R2}$$

The fraction of O($^1$D) reacting with H$_2$O is typically around 10% under tropospheric conditions. It depends on the concentration of water vapour and $L$ and can be calculated from recommended rate constants of reaction (R2) and of competing
quenching reactions (Sander et al., 2011).

Rohrer and Berresheim (2006) reported a linear relationship between OH radical concentrations and $j(\mathrm{O^1D})$ in a five-year data set of measurements at a rural continental site. Even though ozone photolysis is the main primary source of OH via reaction (R2), this result is remarkable because OH concentrations are influenced by a multitude of parameters affecting chemical formation and destruction of OH dependent for example on season, time of day and weather conditions. The result
by Rohrer and Berresheim (2006) can be rationalised by a complex interaction between OH, HO$_2$ and NO$_x$ (= NO + NO$_2$) and compensating seasonal effects of anthropogenic and biogenic OH reactants. Nevertheless, these results highlight the importance of $j(\mathrm{O^1D})$ for the OH concentration and thus for the self-cleaning capability of the atmosphere. Accurate measurements of $j(\mathrm{O^1D})$ are therefore desired.

The magnitude of atmospheric $j(\mathrm{O^1D})$ is determined by the wavelength dependent solar spectral actinic photon flux density
$F_\lambda$:

$$j(\mathrm{O^1D}) = \int \sigma \times \phi \times F_\lambda \, \mathrm{d}\lambda \tag{3}$$

In this equation $\sigma$ and $\phi$ are wavelength and temperature dependent absorption cross sections of ozone and O($^1$D) quantum yields, respectively. $F_\lambda$ is strongly variable and dependent on site-specific and atmospheric parameters, most importantly on solar elevation, clouds, ozone column, aerosol load, altitude and ground albedo.
Techniques for atmospheric measurements of $j(\mathrm{O^1D})$ and other photolysis frequencies were discussed in a review by Hofzumahaus (2006). Absolute techniques are spectroradiometry (spectral measurements of $F_\lambda$ ideally utilising a double-monochromator) and chemical actinometry (chemical change monitoring in suitable reactors). In blind instrument comparisons these methods gave consistent results for $j(\mathrm{O^1D})$ when currently recommended data of $\sigma$ and $\phi$ were used for the calculation according to Eq. (3) (Hofzumahaus et al., 2004). These absolute techniques require complex instrumentation and are therefore
not maintained by many groups. More recently also single-monochromator based spectroradiometers that utilize PDA or CCD arrays were employed for the measurement of photolysis frequencies. However, in particular for $j(\mathrm{O^1D})$ and UV-B measurements in general, these instruments require special care with regard to stray light correction in both laboratory calibrations and



field measurements (Edwards and Monks, 2003; Kanaya et al., 2003; Hofzumahaus et al., 2004; Jäckel et al., 2006; Jäkel et al., 2007; Thiel et al., 2008).

In an earlier radiometric approach Junkermann et al. (1989) developed a filter radiometer for the measurement of $j(O^1D)$. Wavelength separation and detection were made with an interference filter and a solar blind photomultiplier, respectively. This combination had a relative spectral sensitivity similar to the product $\sigma \times \phi$ in Eq. (3). Thus ideally the output of the device was proportional to $j(O^1D)$. For calibration the output signal of the filter radiometer was compared with $j(O^1D)$ from a chemical actinometer. The method worked satisfactory in a limited range of solar zenith angles where the precision of the actinometer was sufficient. However, at $\chi > 60°$ deficiencies of the spectral properties of the filter radiometer were expected to lead to a departure from linearity.

A method to compensate for these deviations was described by Bohn et al. (2004) based on measurements of the spectral sensitivity of a filter radiometer, simulated actinic flux spectra, and the molecular data $\sigma$ and $\phi$ of ozone. Correction factors as a function of total ozone columns ($t_{O_3}$) and solar zenith angles ($\chi$) were derived aiming to linearise the dependence of the output signal on $j(O^1D)$. This correction worked satisfactory at $\chi < 80°$ and $t_{O_3}$ = 315–388 DU for the investigated instrument based on a comparison with a reference spectroradiometer. A similar method is currently applied for most $j(O^1D)$ filter radiometers in use with instrument-specific correction factors.

In a comparison within the European project ACCENT, a number of $j(O^1D)$ filter radiometers were operated simultaneously and compared with a reference spectroradiometer (Bohn et al., 2008). Generally, the results were satisfactory but for some instruments the correction factors were apparently deficient. These correction factors were supplied with the instruments at the time of their purchase in the 1990s but the underlying spectral properties, molecular parameters and simulated spectra were not reported in detail, making it difficult to reproduce the corrections. Moreover, after years of operation the spectral properties of the instruments may have changed and also the recommendations of the O($^1$D) quantum yields changed during the last 20 years (Hofzumahaus et al., 2004). Based on the field comparison alone, no improvement of the parameterised factors was feasible. Therefore, in this work, the spectral sensitivities of six $j(O^1D)$ filter radiometers that took part in the previous comparison were determined in the laboratory and updated correction factors were derived to reevaluate the data. Moreover, to improve the spectral properties of all instruments, interference filters were exchanged, spectral characterisation procedures were repeated and new correction factors were calculated for the modified instruments. Successive field comparisons with a spectroradiometer reference were then consulted to verify the quality of upgraded instruments.

These activities already date back several years and were not reported at the time because the increasing use of detector array spectroradiometers was expected to gradually displace filter radiometers. However, as mentioned above, the determination of $j(O^1D)$ with detector array spectroradiometers is not straightforward and also requires comparisons with reference instruments for validation. Moreover, a number of meantime publications indicates that many filter radiometers are still in use. Therefore an updated account of their performances appears to be appropriate.



## 2 Experimental

### 2.1 Filter radiometers and reference instrument

The $j(O^1D)$ filter radiometers (FR) which took part in this project were similar in construction and were purchased from Meteorologie Consult, Germany. A list with serial numbers is given in Table 1. The instruments came from four institutions within Europe and were in the past deployed in temporary field campaigns as well as for stationary long-term measurements. With one exception (FR 141) the instruments already participated in the ACCENT comparison (Bohn et al., 2008). Two further FZJ instruments of an older series (FR 001 and FR 002) were not operated during ACCENT because of electronic faults but the spectral properties of FR 002 were already determined and approved previously (Bohn et al., 2004).

A technical description of the filter radiometers was given by Junkermann et al. (1989) and Bohn et al. (2004). Basically the instruments are designed for $2\pi$ sr reception of actinic radiation with a quartz-dome diffuser and horizontal shadow ring, an interference filter (Schott, MAZ 8, 300 nm) and a solar-blind photomultiplier (PMT) (Hamamatsu, R-759) for radiation detection. These components are assembled in a water tight housing together with a current amplifier and a heating device. The latter, together with a drying-agent cartridge, prevent condensation of moisture. Via 10–20 m cables the outdoor units were connected to external power and high voltage supplies where the current was converted to a recordable analogue voltage signal (0–5 V). Figure 1 shows a photograph of the compact filter radiometers during a field comparison. Raw data are usually logged with high time resolution to cover rapid changes of $j(O^1D)$ under cloudy conditions. In this work data logging for the field measurements was made with a 16-bit data logger (ADAM 4017) with a 5 s time resolution.

The $j(O^1D)$ reference instrument used in this work was a calibrated spectroradiometer (SR) that was described in detail previously (Hofzumahaus et al., 1999; Bohn et al., 2008). Briefly it is composed of a well characterised quartz receiver (Meterologie Consult GmbH) similar to those used with the filter radiometers, a quartz fibre, a double-monochromator (Bentham DTM 300), and a UV sensitive photomultiplier (EMI). The scanning range was confined to 280–420 nm at a wavelength resolution of 1 nm. The corresponding scanning times were about 120 s. Accuracy of the actinic flux determination is estimated to be 6%. For the calculation of photolysis frequencies the same recommended molecular data of ozone were used as previously (Bohn et al., 2008). Because the time resolution of the spectroradiometer is poorer than that of the filter radiometers, filter radiometer data were averaged over 30 s intervals during which the spectroradiometer was scanning the respective UV-B wavelength range.

### 2.2 Laboratory characterisation

For a quantitative evaluation of the filter radiometer data, the relative spectral sensitivities $D_{rel}$ of the instruments in a range 280-500 nm are required (Sect. 3.2). These sensitivities were determined in the laboratory. However, this was not feasible with the fully assembled instruments for two reasons. Firstly, the transmittance of a typical quartz diffuser is too small for the available tunable laboratory light source. Secondly, the dynamic range of the internal photocurrent amplifier is insufficient for the spectral characterisation where a 5–6 orders of magnitude sensitivity range should be covered. For the laboratory measurements the interference filters (including collimator) and PMTs of the instruments were removed and inserted into a





substitute housing of identical geometric dimensions where the diffuser dome was replaced by a frosted quartz plate. At normal incidence the transmittance of the quartz plate was greater by a factor of about 200 with a minor wavelength dependence ($\pm 5\%$) in the investigated spectral range. Moreover, the substitute allowed direct measurements of the PMT photocurrents with a current-to-voltage amplifier with sufficient dynamic range (Bentham, 228 A) while the high voltages for the PMTs were

produced by the original supply units of the filter radiometers.

The setup of the measurements is depicted in Fig. 2. A high-pressure Xe arc lamp was used as a light source. A mirror reflecting mainly the UV and part of the VIS radiation reduced the thermal load for the components. After passing an optical cut-off filter (Schott, WG 280) the radiation was focussed into a quartz fiber connected to a double-monochromator (Bentham, DTM 300). The use of a double-monochromator ensured negligible stray light and the cut-off filter prevented the detection of

UV radiation in higher orders. A second fiber guided the dispersed radiation from the monochromator exit towards the frosted quartz plate of the filter radiometer substitute. To avoid saturation effects, maximum PMT photocurrents were limited to about 1 $\mu$A. With the selected slit width the monochromatic radiation had a FWHM of about 0.5 nm. The line shape as well as the wavelength positions were checked with a low pressure mercury lamp.

Before and after recording a series of typically 50 spectra with the FR substitutes, a photodiode with known spectral sensitiv-

ity (Hamamatsu, S 1227-1010 BQ) was used as a reference detector to obtain calibrated spectral distributions of the radiation produced by the lamp/monochromator combination (Fig. 2). The ratios of the background corrected, averaged substitute and reference spectra were then normalised to their maxima to obtain the relative spectral sensitivities $D_{\mathrm{rel}}$ of the filter radiometers.

For most instruments PMT hysteresis effects were observed, i.e. after exposure to higher levels of radiation, dark currents were decreased significantly and hardly recovered during the period of a scan ($\approx 2$ min). To avoid the hysteresis effects, the

scanning range was split in two parts which were examined successively intermitted by the photodiode measurements. The first part typically covered a wavelength range 280–325 nm where photocurrents were great enough to be virtually unaffected by hysteresis. The second part of the scanning range was covering a range 320–500 nm. At the beginning of this range photocurrents were much greater than the dark currents but too small to produce any hysteresis. The approach was confirmed by consistent results in the overlapping parts of the two scanning ranges.

## 3 Results and Discussion

### 3.1 Spectral sensitivities

Results of the spectral characterisations are displayed in Fig. 3 where the $D_{\mathrm{rel}}$ of the original instruments are shown in linear and semilogarithmic representations. The upper panel shows similar, narrow peaks at around 299 nm with FWHMs of around 8.5 nm. The lower panel reveals residual sensitivities on the order $10^{-5}$ in a range up to 500 nm. The reason for the residual

sensitivities are insufficient blockings of the interference filters combined with typical sensitivities of the solar blind PMTs of 1–3% in a range 400–500 nm compared to the sensitivities around 300 nm (measured separately for some PMTs after removing the filters). As will be shown in the next section even such small residual $D_{\mathrm{rel}}$ in a range up to 500 nm can affect the performance of the instruments under low sun conditions.





In a first approach to improve the properties of the filter radiometers, an interference filter (Schott, KMD 12) that remained of an older prototype radiometer was inserted into FR 110. This exchange produced a stronger blocking above 350 nm and a somewhat greater FWHM of the transmission curve ($\approx 11$ nm) resulting in an improvement of the instrument performance as intended. Consequently, to be able to fit up more instruments, a batch of new interference filters with a central wavelength of

300 nm, a FWHM of 10 nm and certified blockings $< 5 \times 10^{-6}$ were purchased (Filtrop AG, 300BP10, no. 439100).

The $D_{\rm rel}$ after the exchange of the interference filters are plotted in Fig. 4. Compared to Fig. 3 transmission peaks around 300 nm were about 2 nm broader and sensitivities above 350 nm were typically below $10^{-6}$. Again all $D_{\rm rel}$ exhibited a very similar shape except for FR 110 where the KMD 12 filter was maintained because the transmission peak turned out to be almost ideally situated as will be shown in the next section. As also shown in Fig. 4, the spectral properties with the new interference

filters were very similar to that of FR 002 that was described previously (Bohn et al., 2004) and apparently contained a different interference filter in the first place. Consequently, FR 002 was not modified.

## 3.2 Correction factors

The concept to use the spectral sensitivities $D_{\rm rel}$ and simulated actinic flux density spectra to derive correction factors $f(\chi, t_{\rm O_3})$ dependent on total ozone columns and solar zenith angles was described in detail by Bohn et al. (2004). Ozone columns and

solar zenith angles are the main parameters that determine the shape of the actinic flux density spectra in the wavelength range relevant for O($^1$D) formation ($\approx$300–330 nm). Other atmospheric parameters like aerosol load and clouds may strongly scale the spectra but in a first approximation do not influence their shape in this narrow wavelength range. In addition, ambient temperature influences $j(\mathrm{O^1D})$ because absorption cross sections and quantum yields in Eq. (3) are temperature dependent. However, temperature has virtually no influence on spectral actinic flux densities. The temperature effects can therefore be

separated and accounted for by an additional factor $b(\chi, t_{\rm O_3}, T)$ (Bohn et al., 2004). Consequently, the conversion of filter radiometer voltages $U$ to $j(\mathrm{O^1D})$ is composed of three factors:

$$j(\mathrm{O^1D}) = A_0 \times b(\chi, t_{\rm O_3}, T) \times f(\chi, t_{\rm O_3}) \times U \tag{4}$$

$A_0$ is an absolute calibration factor of dimension $\mathrm{s^{-1}V^{-1}}$ defined for selected reference conditions, in this case $\chi$=30°, $t_{\rm O_3}$=350 DU and $T$= 298 K. The other factors are dimensionless and refer to these reference conditions. Based on the measured spectral

sensitivities, $A_0$ can be determined experimentally using an irradiance standard (Bohn et al., 2004). In addition this requires a characterisation of the angular response of the optical receivers, and specific adjustments caused by the vertical extension of actinic radiation receivers (Hofzumahaus et al., 1999). A second, more direct approach is to determine $A_0$ from parallel measurements with a reference instrument under field conditions. In this work, a spectroradiometer was used as a reference and $A_0$ was obtained from linear regressions of the corresponding $j(\mathrm{O^1D})$ with the products $f(\chi, t_{\rm O_3}) \times U$. Using a spec-

troradiometer has the advantage that temperature effects can be neglected while with a chemical actinometer the gas phase temperature inside the reactor has to be taken into account by the factor $b(\chi, t_{\rm O_3}, T)$ (Hofzumahaus et al., 2004; Bohn et al., 2004). However, $b(\chi, t_{\rm O_3}, T)$ is not instrument specific and merely describes the temperature dependence of $j(\mathrm{O^1D})$ (Bohn et al., 2004). For the instrument characterisations of this work the temperature dependence plays no role and will be neglected





in the following. On the other hand, for applications where the $j(O^1D)$ are used as rate constants in a chemical model, ambient temperatures should be considered. It was checked that the previously evaluated parametrization of $b(\chi, t_{O_3}, T)$ remains valid within about 1% (Bohn et al., 2004). Even though a different reference temperature of 295 K was used, a normalization to any other reference temperature is straightforward.

The factors $f(\chi, t_{O_3})$ were calculated using the experimentally determined $D_{rel}$ and simulated solar actinic flux density spectra according to the following equation:

$$f(\chi, t_{O_3}) = \frac{\int \sigma \times \phi \times F_\lambda \, d\lambda}{\int D_{rel} \times F_\lambda \, d\lambda} \times \frac{\int D_{rel} \times F_\lambda^\circ \, d\lambda}{\int \sigma \times \phi \times F_\lambda^\circ \, d\lambda} \tag{5}$$

Here $\sigma$ and $\phi$ are the molecular ozone data at the selected reference temperature of 298 K that were taken from the literature (Malicet et al., 1995; Matsumi et al., 2002). $F_\lambda^\circ$ refers to a reference spectrum at $\chi=30°$ and $t_{O_3}=350$ DU. All spectra were

simulated for clear sky conditions using the radiation transfer models TUV 4.3 and TUV 5.2 (Madronich and Flocke, 1997) taking the implemented standard ozone concentration profile (scaled to the desired total column), standard clean continental aerosol (AOD=0.235 at 550 nm, SSA=0.99, $\alpha$=1.0), zero ground albedo, zero altitude, and a spectral resolution of 1 nm. The different TUV model versions produced insignificant differences in the correction factors ($<$1%). The same applies for a shift from vacuum wavelengths (TUV standard) to in-air wavelengths at 1000 hPa pressure. For practical reasons, a matrix of

46×26 spectra was utilised covering a range $\chi$=0–90° and $t_{O_3}$=100–600 DU. From these spectra, look-up tables of correction factors were produced for each filter radiometer. Correction factors for actual field conditions were then extracted with a 2D interpolation algorithm using solar zenith angles and ozone columns as input.

    Examples of $f(\chi, t_{O_3})$ at a constant ozone column are shown in Fig. 5. For all filter radiometers the correction factors show a significant variation with solar zenith angle. However, with the new interference filters the factors peak at greater $\chi$ and in

most cases exhibit a weaker dependence compared to the old filters, as intended. For FR 110 an even smaller variation was obtained indicating an almost ideal spectral sensitivity by the use of the KMD 12 filter. Figure 6 shows examples of $f(\chi, t_{O_3})$ at a constant solar zenith angle. For the new interference filters the ozone column dependence is significantly smaller which means that the correction becomes less dependent on the accuracy of the ozone column input. Again for FR 110 the dependence is conveniently weaker and with opposite sign.

In order to understand the differences produced by old and new interference filters, simulated action spectra of ozone photolysis and those of a typical instrument (FR 141) were consulted. Figure 7 shows three examples of spectral photolysis frequencies at different solar zenith angles where the integrals underneath the black curves correspond to $j(O^1D)$. Also plotted are the products $D_{rel}F_\lambda$ for old and new interference filters that were scaled to corresponding integrals. Obviously, the spectral match is improved upon replacement of the interference filter. Moreover, for the largest solar zenith angle, a significant

contribution from a spectral range around 450 nm becomes evident for the old configuration of FR 141. This is caused by the insufficient blocking of the old interference filters and is expected to affect the quality of the corrections when the shapes of natural spectra deviate from those in the simulations. Such deviations are clearly more likely in an extended wavelength range, caused for example by the influence of clouds or aerosols. Obviously, the correction factors can compensate even a significant



spectral mismatch as in the case of the old interference filters. However, the stronger the mismatch, the more dependent the validity of the correction factors become on the agreement of simulated spectra with the actual solar spectra. Figures 5 and 6 indicate that the optimum interference filter would have properties somewhere between the new Filtrop and the re-used Schott KMD 12 with a peak transmittance closer to that of the KMD 12.

Sensitivity tests were made to estimate the influence of various TUV model parameters and atmospheric conditions on the correction factors. The upper and lower panels of Fig. 8 show ratios of correction factors $f_T/f$ as a function of solar zenith angle for old and new interference filters of FR 141, respectively. The $f$ denote the normal correction factors $f(\chi, t_{O_3})$ described in the last paragraph. The $f_T$ were test factors obtained by changing specific parameters in the TUV model identified by three-letter acronyms. In model runs denoted RES the spectral resolution of the actinic flux density spectra was increased

from 1.0 to 0.5 nm. The stronger dependence on spectral resolution for the old configuration is caused by the sharper edges of the sensitivity peak (Fig. 3). Obviously, for the new setup a resolution of 1.0 nm is sufficient. ALT denotes simulations assuming an altitude of 3000 m instead of sea-level, representing a high mountain site. In ALB model runs the ground albedo was increased from 0.0 to 1.0, simulating the maximum possible effect of a fresh snow cover. It should be noted that upward radiation is not considered here, but the effect of increased radiation that is backscattered by the atmosphere is included.

AOD represents polluted conditions with an aerosol optical depth of 1.0 instead of 0.2. Finally, in COD model runs a stratus cloud with an optical thickness of 20 was implemented, corresponding to fairly dimmed overcast conditions. Generally, these modifications had moderate $<10\%$ influence on correction factors and only towards large solar zenith angles. Nevertheless, the advantage of the new interference filters is obvious because variations are significantly reduced. The corrections of the modified instruments should therefore be more robust towards changes of atmospheric conditions during field measurements.

**3.3  Field comparisons**

**3.3.1  Reevaluation of the 2005 ACCENT comparison**

The $j(O^1D)$ filter radiometer results of the previous ACCENT comparison were reproduced in Fig. 9. Ratios of $j(O^1D)$ from filter radiometers and the spectroradiometer reference (SR) were plotted as a function of solar zenith angles. At that time, the correction factors were applied by the participants by their usual evaluation routines. Moreover, the calibration factors $A_0$

determined in the ACCENT comparison were already applied, explaining ratios close to unity at small solar zenith angles. Part of the scatter of the ratios at all solar zenith angles can be attributed to insufficient synchronisation of the measurement techniques, in particular under conditions of broken clouds. However, more systematic deviations at larger $\chi$ were attributed to potential deficiencies of the correction factors (Bohn et al., 2008).

    Figure 10 shows the same data as in Fig. 9 except that the updated correction factors determined in this work were applied

(original instruments, old interference filters). Apart from FR 126 where changes are minor, improvements are apparent in all cases, in particular towards large solar zenith angles, i.e. the performances were significantly improved by the spectral characterisations and the consistent use of molecular ozone data in the calculation of correction factors and of $j(O^1D)$ from the spectroradiometer. The quality of the data of all instruments is very similar now and considered satisfactory. It should be



noted that the type of representation in Figs. 9 and 10 accentuates small absolute differences in particular for $j(O^1D)$ where values go down strongly with increasing solar zenith angle. The color coding indicates that deviations >10% mainly affect data where $j(O^1D)$ is less than 10% of typical noontime summer values at Jülich. Generally, measurements down to solar zenith angles of about 80° are clearly feasible with these instruments. This result is in agreement with a previous evaluation of FR

002 (Bohn et al., 2004). The updated correction factors of the original configurations were transferred to the instrument owners for optional reanalysis of data obtained before the exchange of the interference filters. For convenience, look-up tables and parameterizations of correction factor were supplied.

### 3.3.2  Performance of modified instruments

Ratios of $j(O^1D)$ from the modified filter radiometers and the spectroradiometer are plotted in Fig. 11. The spectral charac-

terisations and successive field comparisons were not made simultaneously for all instruments. Rather there were four measurement periods between spring and autumn where two instruments were processed each. Accordingly, unless the same dates are indicated in the figures, no direct comparison of the data is possible. Nevertheless, it is apparent that all instruments perform satisfactory and exhibit a small scatter towards large solar zenith angels. While compared to Fig. 10 a smaller scatter is expected theoretically as explained in the previous section, the improvement is hard to prove experimentally given differ-

ent measurement periods and weather conditions. At least it is safe to say that there is no deterioration of the performances and that improvements are most likely. Also included in Fig. 11 are results of FR 141 and FR 002 that were not examined during ACCENT but show similar data quality in the successive comparisons. The correction factors and the $A_0$ from the renewed comparisons were again transferred to the instrument owners for the analysis of data obtained after the exchange of the interference filters.

After the modifications in 2006, the different instruments were implemented in various projects, as partly reflected in the publication list in Table 1. All instruments are still in use and repeated recalibrations so far gave no indication that spectral properties change with time. However, care must be taken that housings remain in sound condition and drying agents are replaced regularly. $A_0$ factors typically show slight drifts but neither direction nor magnitude are predictable. For example, in a recalibration in late 2015, filter radiometers FR 120 and FR 126 merely showed $-7\%$ drifts in $A_0$ calibration factors after

almost 10 years of continuous operation at Hohenpeissenberg Observatory. Apparently there is no significant degradation of interference filters or PMT cathodes, even after many years of long term operation. However, it should be noted that the electronics of FR 120 was repaired in mid-2007 which caused a 25% step in signal ratios of FR 120 and FR 126. These ratios were measured regularly at the site with both instruments looking upwards. Afterwards the ratios gradually decreased again and in 2015 accidentally reached same values as obtained in 2006. Of course, $A_0$ drifts can also be caused by fluctuations of

PMT high voltage supplies. It is therefore recommended to monitor or regularly control these high voltages. Moreover, regular comparisons with a reference instrument should be performed for instruments that are used for long-term measurements. Alternatively, a relative drift of $A_0$ can be monitored using suitable, highly stable calibration lamps. For limited field campaigns, calibrations or comparisons before and after the deployment are recommended. Finally, under clear sky conditions with low, assessable aerosol load, also radiative transfer calculations can serve as a reference. At least such comparisons can reveal seri-





ous problems if measured data turn out to be exceptionally low or high. Taking these provisions into account, filter radiometer measurements of $j(O^1D)$ can be classified reliable and of high quality.

## 4 Conclusions

$j(O^1D)$ filter radiometers are instruments with clear advantages regarding convenience and time resolution but require thor-
5    ough characterization and regular calibrations. In this work seven commonly used instruments from the same manufacturer were examined and upgraded. It was shown that spectral sensitivity measurements covering 5-6 orders of magnitude are a prerequisite for the evaluation of reliable correction factors that compensate the dependence of signal outputs on solar zenith angles and ozone columns. Moreover, the quality of the interference filters is important to contain these correction factors in useful ranges. On the other hand, a single spectral characterization is apparently sufficient to derive correction factors that are
10   applicable for many years. Besides these corrections, absolute calibrations that require the availability of a reference instrument like a double-monochromator based spectroradiometer or a chemical actinometer, remain necessary.

*Acknowledgements.* Financial support by the ACCENT project by the European Commission is gratefully acknowledged. The authors thank D. Raak for the electronic fixture of the filter radiometer substitutes, I. Lohse for useful discussions, and S. Madronich for making the TUV radiation transfer model available to the scientific community. Public provision of total ozone column data by the NASA/GSFC TOMS
15   Ozone Processing Team and the TEMIS/ESA team is gratefully acknowledged.



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



**Table 1.** $j(O^1D)$ filter radiometers investigated and modified within this work (instrument number, institution, and references). Abbreviations: UOC (University of Crete, Environmental Chemistry Laboratory), FZJ (Forschungszentrum Jülich, Institut für Energie und Klimaforschung), ULE (University of Leeds, School of Chemistry), DWD (Deutscher Wetterdienst, Meteorologisches Observatorium Hohenpeissenberg).

| Instrument | Institution | References |
|---|---|---|
| FR 102 | UOC | Gerasopoulos et al. (2006); Bohn et al. (2008); Gerasopoulos et al. (2012); Benas et al. (2013) |
| FR 110 | FZJ | Bohn et al. (2006) |
| FR 111 | ULE | Bohn et al. (2008); Hewitt et al. (2010); Whalley et al. (2010); Carpenter et al. (2010); Whalley et al. (2011); Vaughan et al. (2012); Whalley et al. (2015) |
| FR 119 | FZJ | Bohn et al. (2008) |
| FR 120 | DWD | Handisides et al. (2003); Berresheim et al. (2003); Rohrer and Berresheim (2006); Bohn et al. (2008); Decesari et al. (2014) |
| FR 126 | DWD | Handisides et al. (2003); Berresheim et al. (2003); Rohrer and Berresheim (2006); Bohn et al. (2008); Decesari et al. (2014) |
| FR 141[a] | FZJ | Hens et al. (2014); Oswald et al. (2015) |
| FR 002[a,b] | FZJ | Bohn et al. (2004); Bohn et al. (2006); Zhang et al. (2008); Berresheim et al. (2013, 2014) |

[a] Not operative during the ACCENT comparison

[b] Instrument from an older batch, not modified in this work





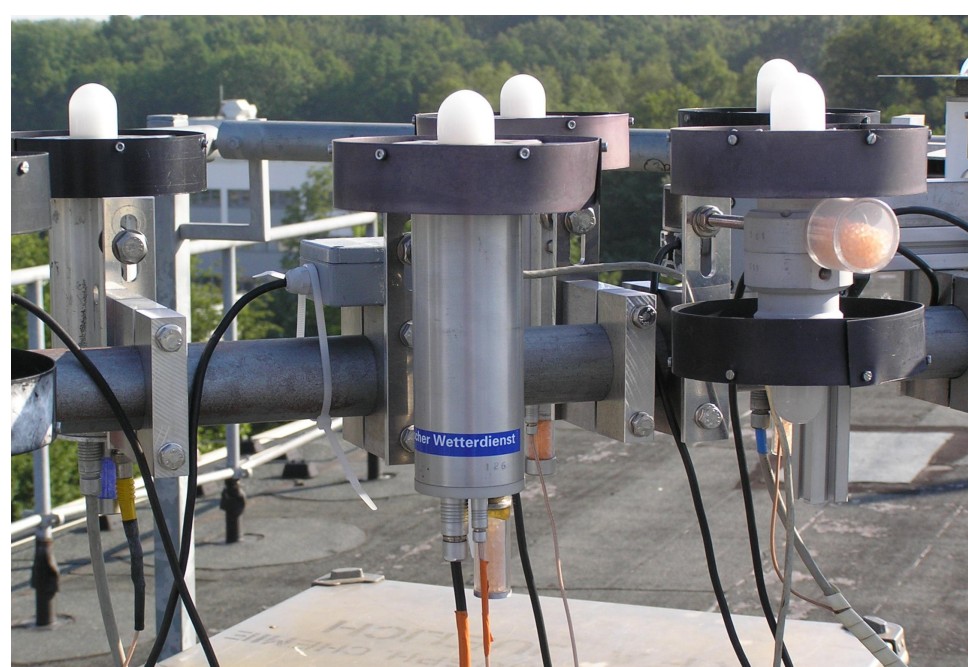

**Figure 1.** Impression of filter radiometers operated side-by-side during the ACCENT intercomparison (Bohn et al., 2008).

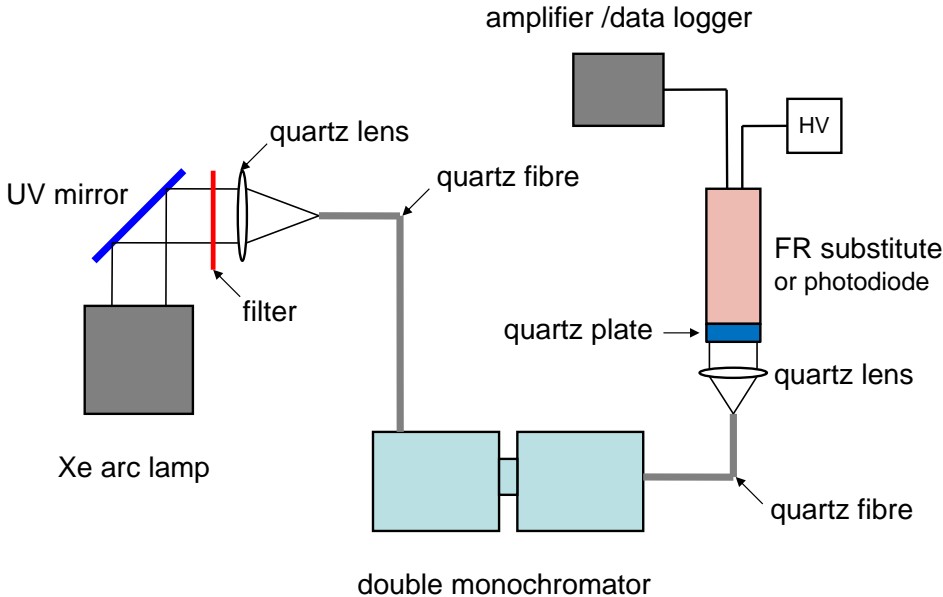

**Figure 2.** Scheme of the laboratory setup for spectral sensitivity measurements.





**Figure 3.** Relative spectral sensitivities $D_{rel}$ of seven $j(O^1D)$ filter radiometers in their original setup. The upper panel shows the main sensitivity peaks on a linear scale. The lower panel depicts an extended wavelength range in a semi-logarithmic plot indicating a residual sensitivity around $10^{-5}$ above 350 nm.





**Figure 4.** Relative spectral sensitivities $D_{\mathrm{rel}}$ as in Fig. 3 after exchange of interference filters: peaks are slightly broader (upper panel) and sensitivities above 350 nm now range around $10^{-6}$ or less because of better blockings (lower panel). For FR 110 an interference filter of a different batch was employed (see text). The dashed curve shows the spectral sensitivity of FR 002 from a previous study (Bohn et al., 2004).



**Figure 5.** Correction factors $f(\chi, t_{O3})$ according to Eq. (5) for all FR instruments as a function of solar zenith angle at a fixed total ozone column of 350 DU. Upper and lower panels show correction factors for old and new interference filters, respectively.





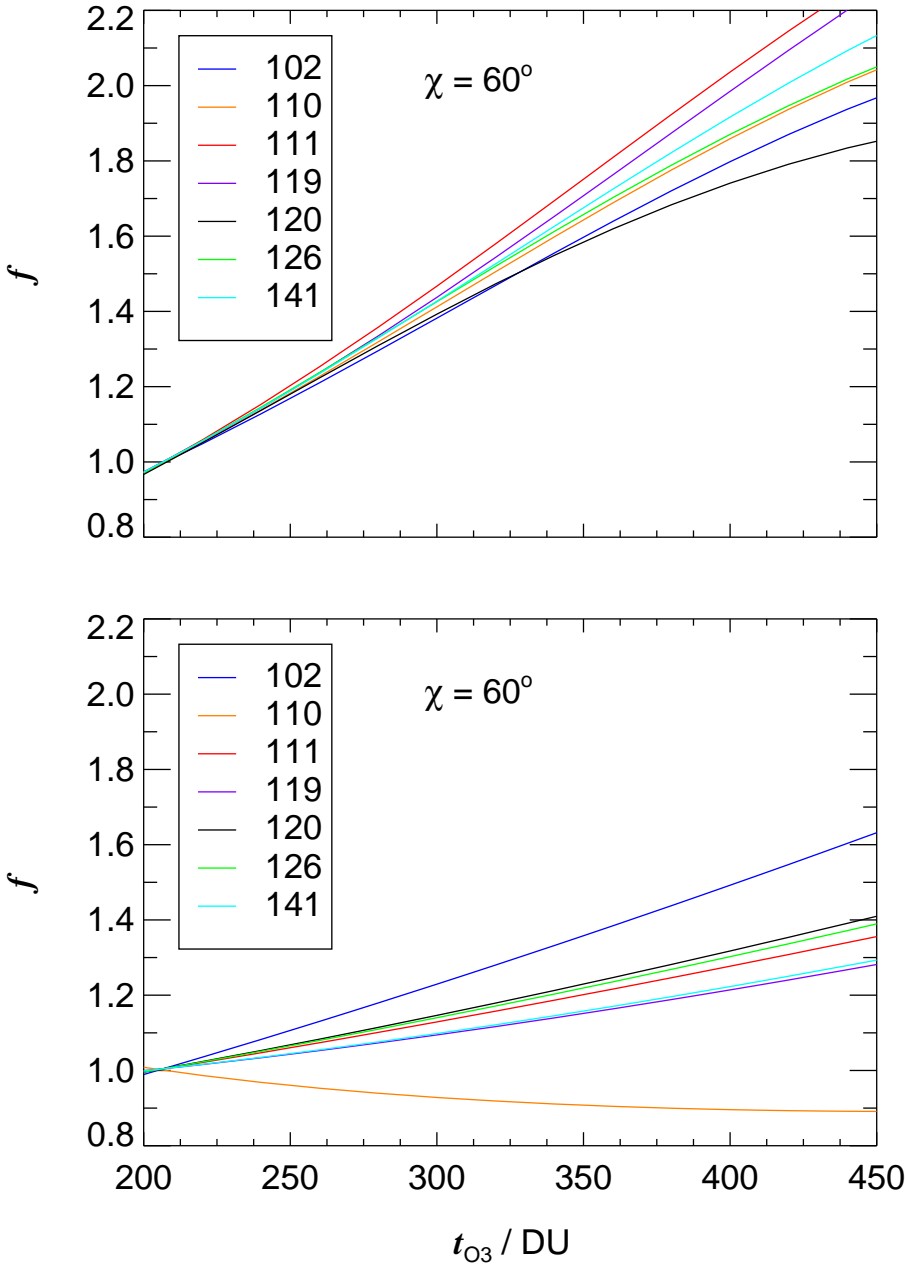

**Figure 6.** Correction factors $f(\chi, t_{O3})$ according to Eq. (5) for all FR instruments as a function of total ozone column at a fixed solar zenith angle of $60°$. Upper and lower panels show correction factors for old and new interference filters, respectively.





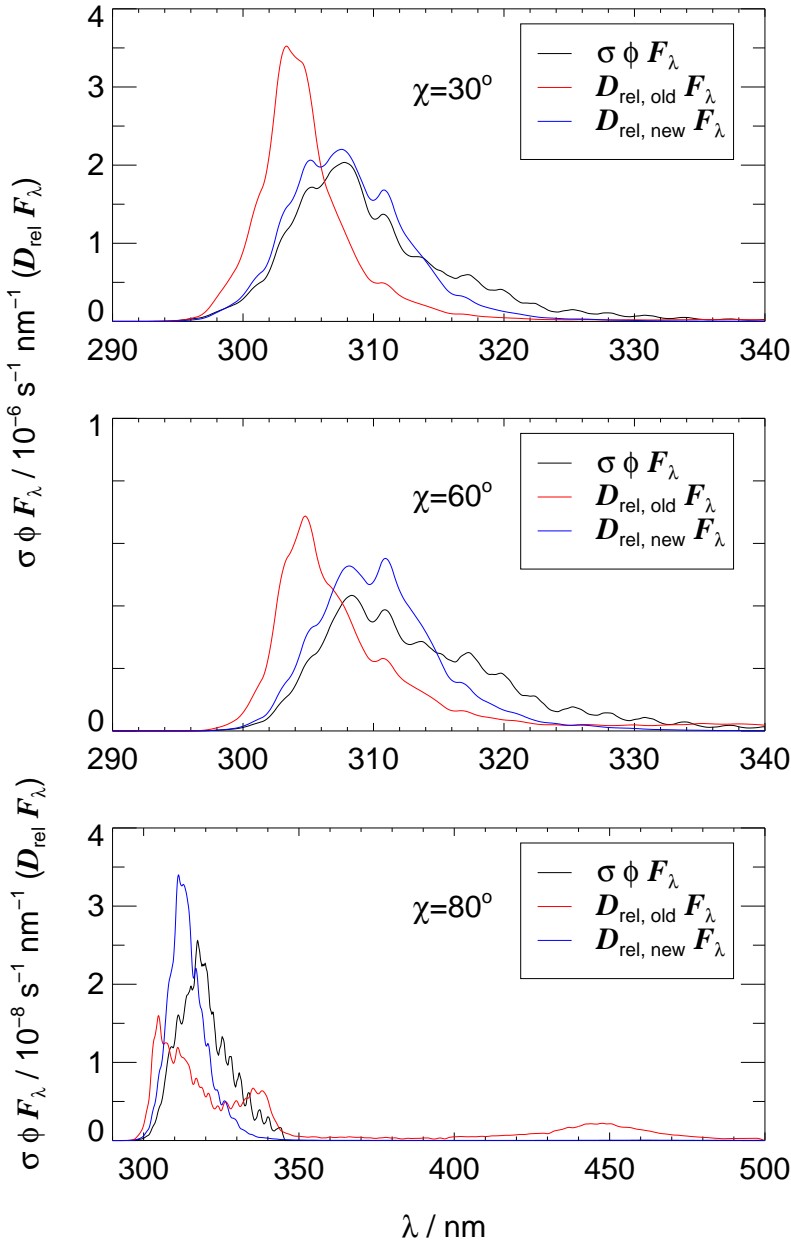

**Figure 7.** Examples of TUV simulated action spectra for three different solar zenith angles of $30°$ (upper panel), $60°$ (middle panel) and $80°$ (lower panel) at a total ozone column of 350 DU. The $j(O^1D)$ correspond to the integrals underneath the black curves. $D_{rel}F_\lambda$ curves, scaled for matching integrals, correspond to old and new spectral sensitivities of FR 141 in red and blue, respectively. Note that the scales in the upper/middle and the lower panel differ by two orders of magnitude and that the wavelength range of the lower panel was extended to indicate the secondary peak in the red curve around 450 nm.





**Figure 8.** Solar zenith angle dependence of ratios $f_T/f$ of correction factors from TUV model runs with specific changes in model parameters. Upper and lower panels show ratios for old and new interference filters, respectively. RES: spectral resolution 0.5 nm / 1 nm. ALT: altitude 3000 m / 0 m. ALB: ground albedo 1.0 / 0.0. AOD: aerosol optical depth 1.0 / 0.24. COD: cloud optical depth 20.0 / 0.0.



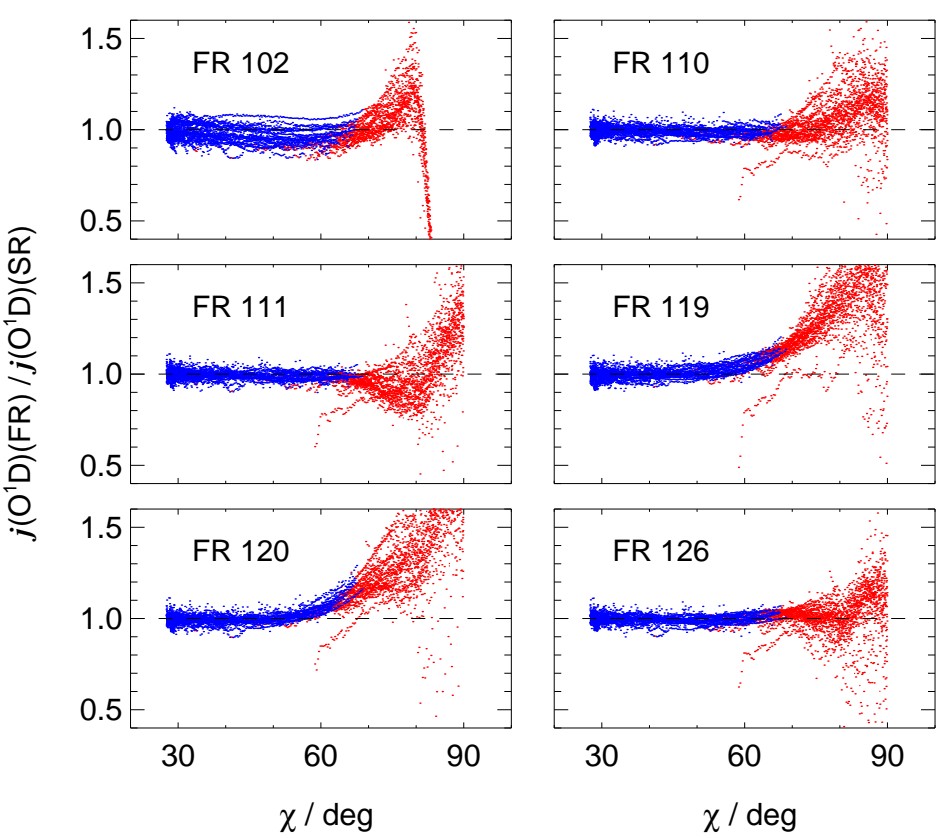

**Figure 9.** Reproduction of a part of Fig. 15 from Bohn et al. (2008) with the $j(O^1D)$ filter radiometer results of the ACCENT comparison. The ratios of $j(O^1D)$ from the filter radiometers and the spectroradiometer reference were plotted as a function of solar zenith angles after application of correction factors $f$ by the participants and of calibration factors $A_0$ from the comparison. Red data points indicate values below 10% of maximum values ($2.8 \times 10^{-5} \, \text{s}^{-1}$). Measurement period: 01-12 June 2005.



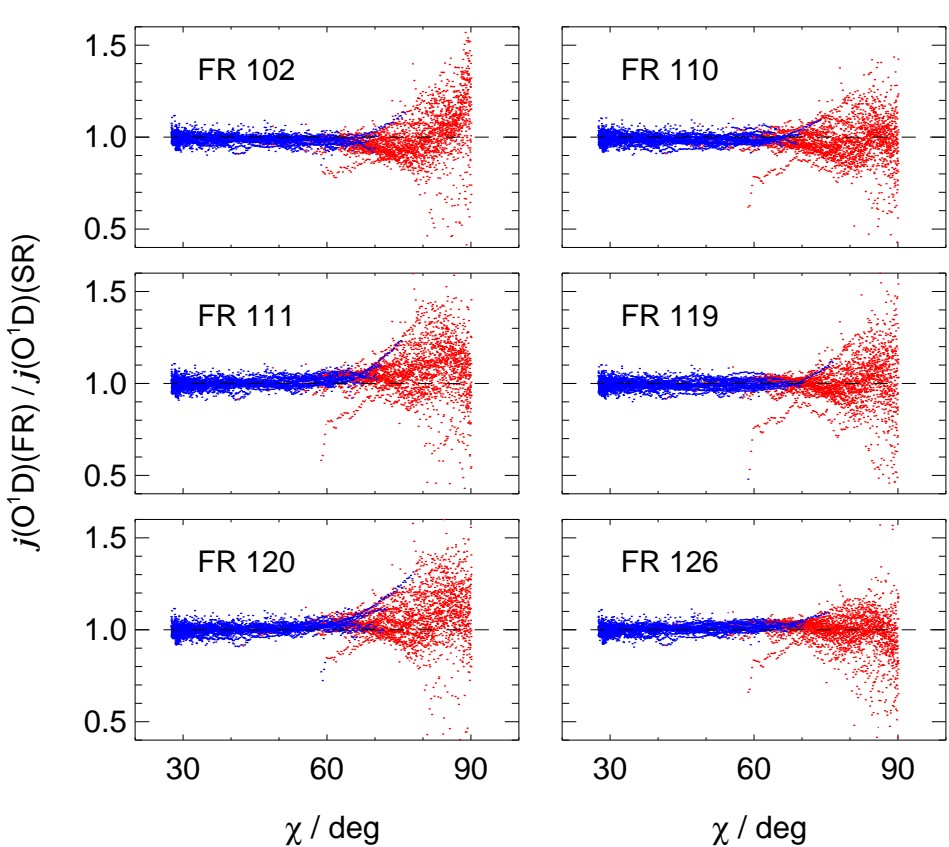

**Figure 10.** Ratios of photolysis frequencies as in Fig. 9 using updated correction factors $f$ based on spectral sensitivities of the original instruments with old interference filters determined in this work.




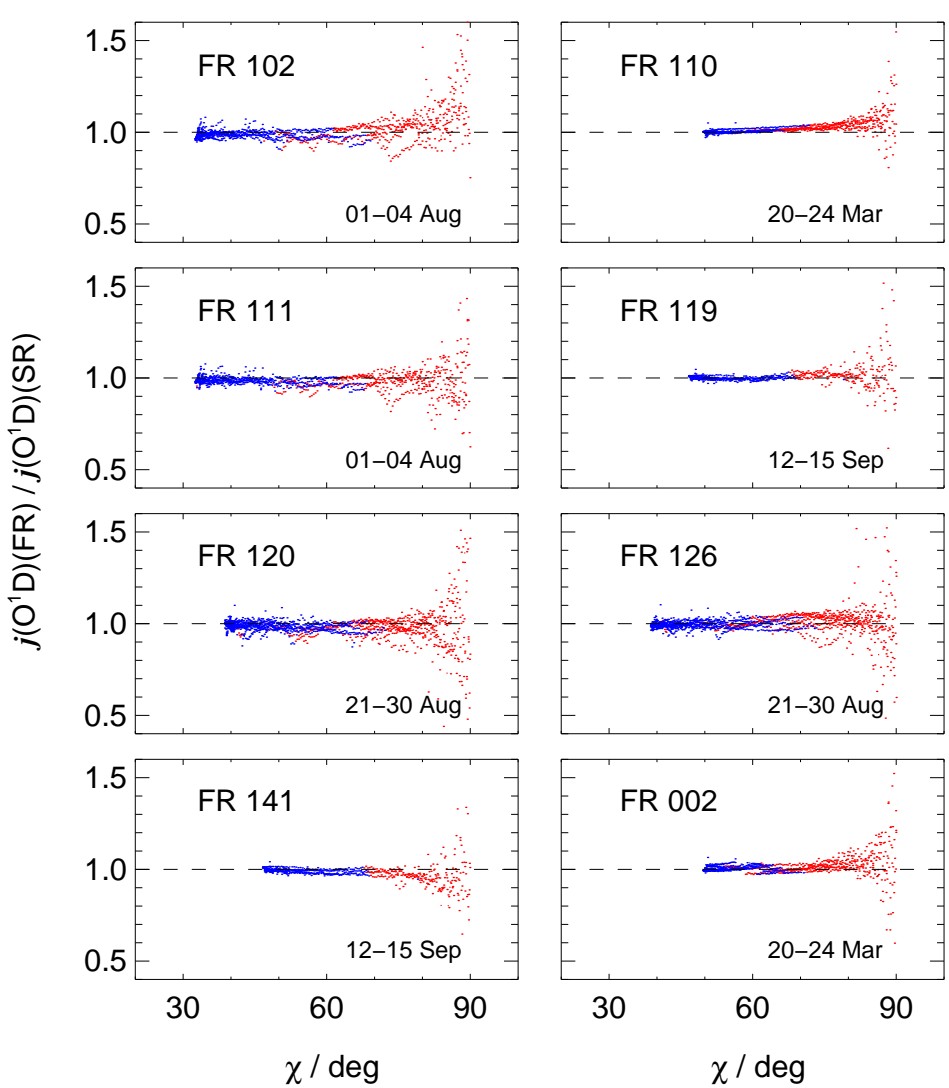

**Figure 11.** Ratios of photolysis frequencies of subsequent comparisons as a function of solar zenith angles after the exchange of interference filters. For direct comparison red data points indicate values below $2.8 \times 10^{-6} \mathrm{s}^{-1}$ as in Fig. 10. Different 3-8 day measurement periods are indicated explaining different ranges of solar zenith angles and number of data points.