# Peer review of "Characterisation and improvement of $j(O^1D)$ filter radiometers"

_Atmospheric Measurement Techniques, 2016_

## Referee Comment (RC1) · Anonymous Referee #1 · 21 Apr 2016

Overall comments.

In this paper, the authors described detailed experiment assessment of the performance of several filter radiometers for the measurement of the photolysis frequency j-(O1D). Although these radiometers have been somewhat superseded by the use of spectroradiometers, as the authors point out, the literature still suggests that the use of these radiometers is still widespread in the atmospheric chemistry community and still provide valuable data on this photolysis frequency especially when spectroradiometer are not available.

The paper describes several important findings about the nature of these radiometers and describes how to employ necessary instrument corrections for the future use of filter radiometers. In addition, which was encouraging to see, the authors re-evaluated

the data from a previous field campaign to ascertain the effect of the new corrections on literature data. The paper is well written and its scope is within the parameters for publication in AMTD following some clarifications I feel would help the reader in understanding the experimental work done here.

Specific comments.

Section 2.2, Following their dismantling, the lab characteristics of the instruments are compared. The authors discuss the parameter Drel and how it was measured from 280-500nm. This "tail" in the sensitivity is discussed in section 3.2 with respect to the potential of counting photons in this region as signal and hence, incorrectly assessing j-(O1D). Although section 3.2 explains this feature, upon first reading it was unclear to me why this spectral region was considered as this is clearly far beyond the normal spectral window of j-(O1D) of 290-340nm. I feel that this section should be reworded or this section merged with section 3.2 to make it clear why such a broad spectral window was evaluated. As this is one of the most important findings of the paper, it would help the reader greatly if this was clarified.

Figure 3: The authors describe the performance of the instruments at the peak and at the tail of the wavelength ranges tested but say nothing about the strange increase in sensitivity (Drel) that is seen in all instruments at around 340nm. I do not understand why all of the radiometers tested show this apparent increase at the traditional wavelength "cut off" for j-(O1D) and this feature should be explained, even if it is removed by the application of the filter described in the paper as shown in Figure 4. Is this some sort of artifact in the PMT response of the instruments?

Conclusion Clearly, the paper describes the improvements made in the determination of j-(O1D) by filter radiometer. As j-(O1D) is the driving force for much of the OH chemistry of the troposphere, it might be useful to include a few sentences on how atmospheric chemists that determined OH concentrations derived from the j-(O1D) data provided by filter radiometer could benefit from the new and improved determinations of

j-(O1D). I assume that the correction factors would be small enough to not significantly affect OH concentrations (which of course, rely on several production and loss steps), but it would be perhaps useful for the authors to add a few sentences on whether they feel literature data of OH should be re-evaluated based on their findings.

Minor comments

Page 4: Line 13: Sentence should read, "The outdoor units were connected to external power via 10-20m cables" rather that the other way around.

Page 8, Line 33: Sentence should read, "The quality of the data is now very similar", rather than the other way around.

---

## Referee Comment (RC2) · Anonymous Referee #2 · 3 Jun 2016

General comments. This work presents a study to characterize several filter radiometers which are used to measured j(O1D). After a characterization in the laboratory and the obtaining of correction factors, measurements from an experimental field have been revaluated. The paper is well written and structured. A sufficient number of references is cited. I suggest to the authors to clarify the goal of this paper. When you read the Introduction and Experimental sections, for me is not clear the objective. It has been necessary to read all the paper to understand the scientific interest. For example you can say: "The goal of this work is to perform a characterization of the several filter radiometers in the laboratory, definition of correction factors and their application, in order to revaluate the experimental data obtained in a field campaign".

Specific comments

[Figure]

Pag. 2. Lines 29-30. "These absolute techniques require complex instrumentation and are therefore not maintained by many groups". I suggest to the authors to remove this sentence.

Pag. 2. Lines 30. "...that utilize PDA or CCD...". Maybe, you can change this sentence as "...that utilize photodiode arrays (PDA) or charge coupled devices (CCD)...".

Pag. 3. Lines 16. "...European project ACCENT,...". Meaning of ACCENT??

Pag. 3. Lines 28-32. I suggest to say in a line and clearly the goal of this work.

Pag. 4. Line 15. "Photograph". Image??

Pag. 4. Line 19. "...previously (Hofzumahaus et al., 1999; Bohn et al., 2008)". I suggest "...previously by Hofzumahaus et al. (1999) and Bohn et al. (2008)".

Pag. 5. Line 6. "A high-pressure Xe arc lamp was used as a light source". What about the characteristics of Xe lamp?

Pag. 5. "Section 3.1. Spectral sensitivities". What temperature was the laboratory during the characterization? You do comments in Pag. 6, lines 32-33.

Pag. 9. Lines 30-34 and Pag. 10. Lines 1-2. I look very interesting the recommendations performed by the authors about the maintenance of the filters radiometers for long-term measurement periods.

Pag. 10. "Conclusions section". In my opinion, more conclusions should be added. This section is very short, only a paragraph.

---

## Author Comment (AC2) · 13 Jul 2016

**Reply to comments by Referee #2**

**General comments.**

This work presents a study to characterize several filter radiometers which are used to measured *j*(O1D). After a characterization in the laboratory and the obtaining of correction factors, measurements from an experimental field have been revaluated. The paper is well written and structured. A sufficient number of references is cited. I suggest to the authors to clarify the goal of this paper. When you read the Introduction and Experimental sections, for me is not clear the objective. It has been necessary to read all the paper to understand the scientific interest. For example you can say: "The goal of this work is to perform a characterization of the several filter radiometers in the laboratory, definition of correction factors and their application, in order to revaluate the experimental data obtained in a field campaign".

Reply: We thank referee #2 for the positive evaluation and give detailed answers to specific questions and recommendations below.

(1) Pag. 2. Lines 29-30. "These absolute techniques require complex instrumentation and are therefore not maintained by many groups". I suggest to the authors to remove this sentence.

Reply: The sentence was removed.

(2) Pag. 2. Lines 30. ": : :that utilize PDA or CCD: : :". Maybe, you can change this sentence as ": : :that utilize photodiode arrays (PDA) or charge coupled devices (CCD): : :".

Reply: The sentence was changed accordingly.

(3) Pag. 3. Lines 16. ": : : European project ACCENT,: : :". Meaning of ACCENT??

Reply: The acronym is explained now (Atmospheric Composition Change - The European Network of Excellence).

(4) Pag. 3. Lines 28-32. I suggest to say in a line and clearly the goal of this work.

Reply: In the second to last paragraph of the introduction we explained the intention of this work in a very similar way as suggested in the referee's general comment. We agree that this information may be missed upon first reading because it is attached to the review of the

results of the previous ACCENT field campaign. In order to make it more clear, we split the paragraph and start the second part with "In this work...":

"In this work, the spectral sensitivities of six  $j(O^1D)$  filter radiometers that took part in the previous ACCENT comparison were determined in the laboratory and updated correction factors were derived to reevaluate the  $j(O^1D)$  field data. Moreover, to improve the spectral properties of all instruments, interference filters were exchanged, spectral characterisation procedures were repeated and new correction factors were calculated for the modified instruments. Successive field comparisons with a spectroradiometer reference were then consulted to verify the quality of upgraded instruments."

(5) Pag. 4. Line 15. "Photograph". Image??

Reply: We use "image" now.

(6) Pag. 4. Line 19. ": : :previously (Hofzumahaus et al., 1999; Bohn et al., 2008)". I suggest ": : :previously by Hofzumahaus et al. (1999) and Bohn et al. (2008)".

Reply: Changed as recommended.

(7) Pag. 5. Line 6. "A high-pressure Xe arc lamp was used as a light source". What about the characteristics of Xe lamp?

Reply: We added a sentence: "This type of lamp emits a high-intensity, almost continuous spectrum in a range 200-1200 nm."

(8) Pag. 5. "Section 3.1. Spectral sensitivities". What temperature was the laboratory during the characterization? You do comments in Pag. 6, lines 32-33.

Reply: Temperature in the laboratory during the characterisations was around normal room temperature ( $\approx 22^{\circ}$ ) but it plays no role. For the interference filters a very low temperature response is specified. In separate experiments it was tested that the slight heating ( $\approx 30^{\circ}$ ) that is performed to prevent condensation of moisture in the outdoor units, does not influence the PMT response. Therefore no significant temperature effect on the radiation measurements is expected. The remark on page 6 refers to the temperature dependence of *j*(O1D) that results from the temperature dependence of O(1D) quantum yields and of O3 absorption cross

sections. As explained in the text this temperature effect is completely separable from the measurements.

(9) Pag. 9. Lines 30-34 and Pag. 10. Lines 1-2. I look very interesting the recommendations performed by the authors about the maintenance of the filters radiometers for long-term measurement periods.

Reply: We gave these instructions because there are various options users may consider to ensure data quality.

(10) Pag. 10. "Conclusions section". In my opinion, more conclusions should be added. This section is very short, only a paragraph.

Reply: We extended the conclusions section which is also in line with suggestions by Referee 1:

"These calibrations ensure that the measured data are accurate, in particular under conditions of small solar zenith angles when  $j(O^1D)$  is high and important, e.g. for predictions of noontime OH radical concentrations and the atmospheric oxidizing capacity. The complementary correction factors gain significance under conditions with low sun when  $j(O^1D)$  is getting smaller which is important, e.g. for an accurate assessment of ozone photolysis compared to other primary radical sources like HNO2 or ClNO2 photolysis in the early morning. Overall, filter radiometers are suitable to accurately measure  $j(O^1D)$  in a wide dynamic range. In this work previously described deficiencies of the investigated instruments were examined and widely removed. However, these deficiencies are considered moderate and require no major revision of previous work caused by incorrect  $j(O^1D)$ ."

---

## Author Comment (AC1)

**Reply to comments by Referee #1**

*Overall comments.*

*In this paper, the authors described detailed experiment assessment of the performance of several filter radiometers for the measurement of the photolysis frequency j-(O1D). Although these radiometers have been somewhat superseded by the use of spectroradiometers, as the authors point out, the literature still suggests that the use of these radiometers is still widespread in the atmospheric chemistry community and still provide valuable data on this photolysis frequency especially when spectroradiometer are not available.*

*The paper describes several important findings about the nature of these radiometers and describes how to employ necessary instrument corrections for the future use of filter radiometers. In addition, which was encouraging to see, the authors re-evaluated the data from a previous field campaign to ascertain the effect of the new corrections on literature data. The paper is well written and its scope is within the parameters for publication in AMTD following some clarifications I feel would help the reader in understanding the experimental work done here.*

We thank referee #1 for the positive evaluation and give detailed answers to specific questions and recommendations below:

*Specific comments.*

(1) *Section 2.2, Following their dismantling, the lab characteristics of the instruments are compared. The authors discuss the parameter Drel and how it was measured from 280 500nm. This "tail" in the sensitivity is discussed in section 3.2 with respect to the potential of counting photons in this region as signal and hence, incorrectly assessing j-(O1D). Although section 3.2 explains this feature, upon first reading it was unclear to me why this spectral region was considered as this is clearly far beyond the normal spectral window of j-(O1D) of 290-340nm. I feel that this section should be reworded or this section merged with section 3.2 to make it clear why such a broad spectral window was evaluated. As this is one of the most important findings of the paper, it would help the reader greatly if this was clarified.*

Reply: We agree that the wide spectral range considered may confuse readers upon first reading. We added a sentence that gives an explanation at the beginning of Sect. 22:

"For a quantitative evaluation of the filter radiometer data, the relative spectral sensitivities $D_{rel}$ of the instruments in a range 280-500 nm are required. This wide spectral range is necessary because of an imperfect blocking of interference filters resulting in unwanted

signal contributions, as explained in more detail in Sects. 3.1 and 3.2. The spectral sensitivities were determined ...”

(2) *Figure 3: The authors describe the performance of the instruments at the peak and at the tail of the wavelength ranges tested but say nothing about the strange increase in sensitivity (Drel) that is seen in all instruments at around 340nm. I do not understand why all of the radiometers tested show this apparent increase at the traditional wavelength "cut off" for j-(O1D) and this feature should be explained, even if it is removed by the application of the filter described in the paper as shown in Figure 4. Is this some sort of artifact in the PMT response of the instruments?*

Reply: This spectral feature comes from the type of interference filter used in the old configuration. It also shows up in the manufacturer's data sheets. We'll mention this at the end of the first paragraph of Sect. 3.1:

"As will be shown in the next section even such small residual $D_{rel}$ in a range up to 500 nm can affect the performance of the instruments under low sun conditions. The secondary peak around 340 nm found for all instruments is a feature of the MAZ-8 interference filter which is in line with the typical transmittance curve provided by the manufacturer."

(3) *Conclusion Clearly, the paper describes the improvements made in the determination of j-(O1D) by filter radiometer. As j-(O1D) is the driving force for much of the OH chemistry of the troposphere, it might be useful to include a few sentences on how atmospheric chemists that determined OH concentrations derived from the j-(O1D) data provided by filter radiometer could benefit from the new and improved determinations of j-(O1D). I assume that the correction factors would be small enough to not significantly affect OH concentrations (which of course, rely on several production and loss steps), but it would be perhaps useful for the authors to add a few sentences on whether they feel literature data of OH should be re-evaluated based on their findings.*

Reply:
We extended the conclusions section accordingly and added a few sentences to clarify the importance of calibrations and correction factors for radical chemistry related questions:
"These calibrations ensure that the measured data are accurate, in particular under conditions of small solar zenith angles when $j(O^1D)$ is high and important, e.g. for predictions of noontime OH radical concentrations and the atmospheric oxidizing capacity. The complementary correction factors gain significance under conditions with low sun when $j(O^1D)$ is getting smaller which is important, e.g. for an accurate assessment of ozone

photolysis compared to other primary radical sources like $HNO_2$ or $ClNO_2$ photolysis in the early morning. Overall, filter radiometers are suitable to accurately measure $j(O^1D)$ in a wide dynamic range. In this work previously described deficiencies of the investigated instruments were examined and widely removed. However, these deficiencies are considered moderate and require no major revision of previous work caused by incorrect $j(O^1D)$."

*(4) Minor comments*

*Page 4: Line 13: Sentence should read, "The outdoor units were connected to external power via 10-20m cables" rather that the other way around.*
*Page 8, Line 33: Sentence should read, "The quality of the data is now very similar", rather than the other way around.*

Reply: These changes were made as recommended.